# High Refractive Index Silica-Titania Films Fabricated via the Sol–Gel Method and Dip-Coating Technique—Physical and Chemical Characterization

**DOI:** 10.3390/ma14237125

**Published:** 2021-11-23

**Authors:** Magdalena Zięba, Katarzyna Wojtasik, Cuma Tyszkiewicz, Ewa Gondek, Jacek Nizioł, Katarzyna Suchanek, Michał Wojtasik, Wojciech Pakieła, Paweł Karasiński

**Affiliations:** 1Department of Optoelectronics, Silesian University of Technology, B. Krzywoustego 2, 44-100 Gliwice, Poland; magdalena.zieba@polsl.pl (M.Z.); cuma.tyszkiewicz@polsl.pl (C.T.); 2Department of Physics, Cracow University of Technology, Podchorążych 1, 30-084 Kraków, Poland; egondek@pk.edu.pl (E.G.); katarzyna.suchanek@pk.edu.pl (K.S.); 3Faculty of Physics and Applied Computer Science, AGH University of Science and Technology, al. Mickiewicza 30, 30-059 Krakow, Poland; niziol@agh.edu.pl; 4Oil and Gas Institute—National Research Institute, Lubicz 25A, 31-503 Krakow, Poland; wojtasik@inig.pl; 5Faculty of Mechanical Engineering, Institute of Engineering Materials and Biomaterials, Silesian University of Technology, ul. Konarskiego 18a, 44-100 Gliwice, Poland; wojciech.pakiela@polsl.pl

**Keywords:** silica–titania films, sol–gel, dip-coating technique, optical waveguide film, optical band gap, optical losses, planar waveguide

## Abstract

Crack-free binary SiO_x_:TiO_y_ composite films with the refractive index of ~1.94 at wavelength 632.8 nm were fabricated on soda-lime glass substrates, using the sol–gel method and dip-coating technique. With the use of transmission spectrophotometry and Tauc method, the energy of the optical band gap of 3.6 eV and 4.0 eV were determined for indirect and direct optical allowed transitions, respectively. Using the reflectance spectrophotometry method, optical homogeneity of SiO_x_:TiO_y_ composite films was confirmed. The complex refractive index determined by spectroscopic ellipsometry confirmed good transmission properties of the developed SiO_x_:TiO_y_ films in the Vis-NIR spectral range. The surface morphology of the SiO_x_:TiO_y_ films by atomic force microscopy (AFM) and scanning electron microscopy (SEM) methods demonstrated their high smoothness, with the root mean square roughness at the level of ~0.15 nm. Fourier-transform infrared (FTIR) spectroscopy and Raman spectroscopy were used to investigate the chemical properties of the SiO_x_:TiO_y_ material. The developed binary composite films SiO_x_:TiO_y_ demonstrate good waveguide properties, for which optical losses of 1.1 dB/cm and 2.7 dB/cm were determined, for fundamental TM_0_ and TE_0_ modes, respectively.

## 1. Introduction

Thin film technologies have played a key role in the development of microelectronics, and now they play the same role in optoelectronics. Attractive materials for the application in optoelectronics include transition metal oxides films (TMOs) (TiO_2_, ZnO, HfO_2_, ZrO_2_, Ta_2_O_5_) which, due to high optical energy band gaps, show good transmission properties in the Vis–NIR spectral range [1,2,3,4,5,6,7,8]. The TiO_2_ and ZnO films have sensor [9,10,11,12] and photocatalytic properties [13,14]. In the structures of photovoltaic cells, they are used as buffer films [15] or as acceptor films, blocking holes and effectively returning electrons to the anode [16,17]. TMO films with high refractive index and silica films (SiO_2_) with low refractive index are used as components of multilayer photonic structures, such as dielectric mirrors [18,19] or anti-reflective structures [20,21] for applications in photovoltaics. TMOs are also used in integrated optics technologies [22,23,24,25,26,27,28]. TMO films can be produced using physical vapor deposition (PVD) methods, including e-beam evaporation [2,4], magnetron sputtering [7,8,9,13,23,29], pulse laser deposition PLD [30], or chemical methods, including chemical spray pyrolysis [12], atomic layer deposition (ALD) [5,10,14,17] and the sol–gel method [6,18,20,22,26,28]. PVD methods require utilization of specialized technological instruments including vacuum systems. They are excellent for microelectronics applications; however, their yield in optoelectronics applications is low. Furthermore, the smoothness of TMO films fabricated using PVD methods is too low for application in integrated optics. A particularly attractive fabrication method of TMO films is the sol–gel method, whereof the most important advantages involve the ability to control the structure of material in a wide range as well as its high efficiency. The sol–gel method does not require complex technological installation, which is a significant advantage in terms of its implementation in small and medium-sized companies. The TMO films tend to crystallize, which is not an advantage in terms of optical applications as this leads to a dramatic increase in light scattering. For example: the crystallization of pure TiO_2_ started at ~200 °C [31]. Additionally, the reduction in the annealing temperature of films fabricated with the sol–gel method below 430 °C does not remove all organic residues from them. The crystallization of TMO can be weakened by adding silicon dioxide (SiO_2_) [31,32,33]. Thus, amorphous films can be obtained by the development of binary systems MO: SiO_2_, where MO is a metal oxide. Consequently, by combining silica of low refractive index and MO of high refractive index, the possibility of shaping the refractive index in the range from ~1.45 to over 2 emerges. The greatest possibilities in terms of shaping the refractive index are provided by the use of titanium dioxide (*n_TiO_*_2_~2.5) with silicon dioxide (*n_SiO_*_2_~1.45). Sample applications of thin films fabricated using the sol–gel method are presented in Refs [34,35,36,37].

The subject of this work involves composite films SiO_x_:TiO_y_ of high refractive index, fabricated by the sol–gel method and dip-coating technique. Binary composite films SiO_x_:TiO_y_ with the proportion of components SiO_x_:TiO_y_ = 1:1 have been investigated by us for many years [38,39,40,41,42]. The films developed by us have high refractive index *n*~1.8, low optical losses, they remain stable over many years [40] and are characterized by high chemical and optical homogeneity [41]. These films are the material platform on which we develop the structures of integrated optics for applications in chemical/biochemical evanescent wave sensors [38,39]. On the basis of these waveguide films, grating couplers [38] and rib optical waveguides as well as directional couplers [39] have been developed. In this work, we present the results of our research on SiO_x_:TiO_y_ composite films with even higher refractive indexes (*n*~1.94 at wavelength 632.8 nm), also fabricated by the sol–gel method and dip-coating technique. In the waveguide films of the refractive index of *n*~1.91 at wavelength 632.8 nm, we produced grating couplers whereof sensor properties were the subject of our earlier work [42]. The optical properties of the SiO_x_:TiO_y_ composite waveguide films presented here were investigated using spectrophotometric and ellipsometric methods. Surface morphology was investigated using the atomic force microscopy (AFM) and scanning electron microscopy (SEM), and chemical properties were tested using the methods of Fourier transform infrared (FTIR) and Raman spectroscopies. We also present the research results involving waveguide properties of the developed SiO_x_:TiO_y_ composite films.

The work is organized as follows. Section 2 describes the basics of the sol–gel method and the dip-coating technique, preparation method of the sol and the fabrication method of SiO_x_:TiO_y_ composite films, the applied materials used and technological procedures. Section 3 describes the applied measurement methods and apparatus used to characterize the presented composite films. The obtained results and their discussion are presented in Section 4, which also demonstrates that the developed SiO_x_:TiO_y_ composite films exhibit good waveguide properties.

## 2. Technology

### 2.1. Fundamentals

Binary composite films SiO_x_:TiO_y_ were fabricated using the sol–gel method and dip-coating technique. The fabrication processes of films in this way have multi-stage character, as illustrated in Figure 1.

At the first stage, a sol is formed, which is the result of the hydrolysis of precursors. Almost simultaneously, the condensation process of hydrolysis products begins. In the hydrolysis reaction, the alkoxy groups are replaced with hydroxyl groups, according to reactions [43,44,45]:(1)$$M{\left(\mathrm{OR}\right)}_{4}{+H}_{2}O\to \mathrm{HO}-M{\left(\mathrm{OR}\right)}_{3}+\mathrm{ROH}$$
or
(2)$$M{\left(\mathrm{OR}\right)}_{4}{+4H}_{2}O\to M{\left(\mathrm{OH}\right)}_{4}+4\mathrm{ROH}$$
where: M—metal or non-metal atom (in our case Si or Ti), R—alkyl group, ROH—alcohol.

The hydrolysis process takes place partially or completely, depending on the amount of solvent. The condensation reaction, which takes place between two hydroxyl groups or one hydroxyl and alkoxyl group, leads to the formation of bonds between a metal or non-metal atom and oxygen [44,45]:(3)$${\left(\mathrm{OR}\right)}_{3}M-\mathrm{OH}+\mathrm{OH}-M{\left(\mathrm{OR}\right)}_{3}\to {\left(\mathrm{OR}\right)}_{3}M-O-M{\left(\mathrm{OR}\right)}_{3}{+H}_{2}O$$
or
(4)$${\left(\mathrm{OR}\right)}_{3}M-\mathrm{OH}+\mathrm{OH}-M{\left(\mathrm{OR}\right)}_{3}\to {\left(\mathrm{OR}\right)}_{3}M-O-M{\left(\mathrm{OR}\right)}_{3}+\mathrm{ROH}$$

These reactions can be catalyzed by acids or bases.

There are three techniques used to production of thin layers in the sol–gel method [44]: spin-coating, dip-coating, and the meniscus coating technique. In this paper, the dip-coating technique was applied for the production of the SiO_x_:TiO_y_ waveguide layers on the glass substrate. This technique consists of several steps, which are presented in Figure 2. In the first stage, the glass substrate is immersed in the sol. After dipping the glass substrate in the sol, the glass substrate with a thin layer of sol is pulled up at a constant and controlled speed. After the removal of the glass substrate from the sol solution, a homogeneous film is formed on the substrate’s surface. The sol films deposited on the substrate are then subject to drying and annealing. The thickness of the layer depends on the viscosity of the sol *η*, the liquid–vapor surface tension *σ**,* and especially the withdrawal speed *v* of the glass substate from the sol [46]. The general dependence of the film thickness *d* on substrate withdrawal speed *v* can be written in the following form [47]:(5)$$d=A\xi v^{\alpha }$$
where *ξ* = 1 (cm/min)^−^^α^ is the scaling factor. Depending on the properties of the sol and substrate withdrawal speed, its movement may cause a change in the curvature of sol meniscus at the boundary sol/substrate or not. When the sol exhibits the properties of a Newtonian fluid, then in the first case the slope *α* = 1/2 and the proportionality coefficient is [44,46]:(6)$$A=c_{1}{\left(\frac{\eta }{\rho g}\right)}^{1/2}$$
where *c*_1_ ≈ 0.8, *ρ* is the density of sol, and *g* is the gravity acceleration. Additionally, in the second case, slope *α* = 2/3 and the proportionality coefficient determined by Landau and Levich [44,46] is:(7)$$A=\frac{0.94\eta^{2/3}}{\sigma^{1/3}{\left(\rho g\right)}^{1/2}}$$
where: *η*—surface stress along the phase border sol–air, *σ*—viscosity of sol.

In the above equations, the evaporation process of solvents was not taken into account. It should be emphasized that the thicknesses of the films in the drying and annealing processes undergo contraction and they are generally much smaller than the thicknesses of sol films immediately after their application [48]. In practice, the proportionality coefficient *A* and slope *α* change with the aging time of the sol [47]. These coefficients are determined from the approximation of experimental dependencies *d* = *d*(*v*) for the films after annealing, and they are used to design technological processes. In Section 4 we will demonstrate that the presented here composite films SiO_x_:TiO_y_ have good waveguide properties. From the optical point of view, after coating the substrate with a composite film of higher refractive index than that of the substrate, a three-film optical waveguide structure is formed on both sides of the substrate whereof the diagram is presented in Figure 3.

### 2.2. Materials

Tetraethyl orthosilicate (TEOS) and titania (IV) ethoxide (TET) were used as precursors of silica, and titania, respectively. Both precursors were purchased from Sigma-Aldrich (Steinheim, Germany) and used without any further purification. Anhydrous ethanol (EtOH; 99.8%) and hydrochloric acid (HCl; 36%) were supplied by Avantor Performance Materials (Gliwice, Poland). Soda-lime microscope slides (Menzel Gläser, Thermo Scientific, *n* = 1.513 at 632.8 nm) of dimension 76 × 26 × 1 mm^3^ were used as substrates. Deionized water was used directly from the deionizer (Polwater DL2-100S613TUV, Labopol Solutions&Technologies, Kraków, Poland). The final SiO_x_:TiO_y_ synthesized sol was filtered through a 0.1 μm PTFE syringe filter (Puradisc 25 TF, Whatman, Maidstone, UK).

### 2.3. Sol Preparation and Film Fabrication

Silica–titania (SiO_x_:TiO_y_) waveguide films were prepared via the sol–gel method and dip-coating technique, which was described in Section 2.1 and was presented in Figure 1 and Figure 2. The preparation of gel solution was carried out in two stages. For the preparation of sols, we used ethanol as a homogenizing agent and hydrochloric acid (36%) as the catalyst. In the first stage, two solutions of sols were separately prepared. For solution A, 0.09 mole of TEOS was mixed with 0.45 moles of ethanol, 0.18 moles of deionized water and 0.006 mole of HCl. For solution B, 0.17 mole of TET was mixed with 1.4 moles of ethanol, 0.3 moles of deionized water and 0.08 moles of HCl. Then, the solution B was added to the partially hydrolyzed solution A, and the formation process of the gel solution was continued in an ultrasound field for 6 h at a temperature of 50 °C. The final molar ratio of (TET+TEOS): EtOH:H_2_O:HCl was 1:7.2:1.8:0.3, whereas the molar ratio of titania to silica was 1.94:1. Glass substrates were cleaned mechanically using a detergent solution, soaked in a solution of acetic acid, then isopropanol, and finally acetone. After each step (without the last one), the substrates were rinsed with distilled water. After filtering, the sol was stored at a temperature of 4 °C. Composite SiO_x_:TiO_y_ films were deposited on the glass substrates dipped vertically in sol and withdrawn in a range of speed of 3.6—7.0 cm/min. After deposition by dip-coating technique, the films were annealed at 500 °C for 1h under atmospheric pressure. Quality, and in particular uniformity, of films deposited using the dip-coating technique are sensitive to conditions in which those films are deposited. The issue of the effect the volume of sol and vessel as well as the atmosphere in which dip-coating processes are carried out are discussed in Ref. [49].

For FTIR measurements, a sample of the sol (~3 mL) was dried and annealed at 500 ° C for 1 h, yielding dry powder. For the purposes of measurements, using spectroscopic ellipsometry, a SiO_x_:TiO_y_ film was fabricated on the silicon substrate.

## 3. Methods and Materials

The produced films and the sol were subjected to physical and chemical characterization. The physical characterization consisted of examining the optical properties and surface morphology of the produced films, and the chemical characterization consisted of determining the chemical composition of the sol and that of the synthesized material in solid state. The surface morphology of the films was examined using the AFM and SEM measurement methods. The optical characterization was performed using spectrophotometric and ellipsometric methods. The chemical characterization was performed using the FTIR and Raman spectroscopies.

### 3.1. Fourier Transform Infrared Spectroscopy

The analysis of FTIR spectra recorded for the sol and for the powder obtained from the same sol allowed for the identification of chemical bonds in the materials of these samples. FTIR spectra were acquired using a Thermo Nicolet iS5 infrared spectrometer equipped with an ATR accessory. Spectra were performed at the resolution of 4 cm^−1^ in the range of 4000–400 cm^−1^, and 64 scans.

### 3.2. Raman Spectroscopy

The Raman spectrometer used in the study was a high resolution, confocal micro-spectrometer Almega XR of Thermo Electron Corp. The chosen excitation light wavelength was 532 nm, objective magnification 100×, and a pinhole aperture of 25 μm. Data were recorded in the spectral range from approx. 100 cm^−1^ up to 4000 cm^−1^ and with the spectral resolution of 2 cm^−1^.

### 3.3. Spectrophotometry

Transmittance and reflectance spectra of the SiO_x_:TiO_y_ thin films were used to determine of transmission properties, optical homogeneity and optical band gaps. The energies optical band gap of the SiO_x_:TiO_y_ films were determined by plotting Tauc’s equations, which shows the dependence between absorption coefficient and photon energy:(8)$$\alpha \cdot h\nu =B{\left(h\nu -E_{g}\right)}^{r}$$
where: *B* is a absorption constant, which does not depend on photon energy *h**ν*, *E*_g_ is the band gap of the material, *v* is the frequency of the incident radiation, *h* is the Planck’s constant and power coefficient *r* is the power coefficient, of which the value determines the optical transition. The coefficient *r* takes the value of 2 for indirect allowed optical transitions and the value of ½ for direct allowed optical transitions.

Based on the reflection spectra, the optical homogeneity of SiO_x_:TiO_y_ composite films was determined. For this purpose, the reflection spectra of the film and those of the substrate were compared. When the film is optically homogeneous, then in the spectral range away from the absorption edge, the interference minima of the reflection spectrum of the film lie on the reflection characteristics of the substrate [41,42].

Transmittance and reflectance studies of the SiO_x_:TiO_y_ waveguide films were carried out by using a UV–Vis AvaSpec-ULS2048LTEC Spectrophotometer (Avantes, Apeldoorn, The Netherlands). The AvaLight-DH-S-BAL (Avantes) was used as a light source. The spectra were recorded in the wavelength range of 200–1100 nm, at room temperature.

### 3.4. Ellipsometry

The basic ellipsometry equation has the following form [50]:(9)$$\rho =\frac{r_{p}}{r_{s}}=\tan\Psi \cdot \exp(i\Delta )$$
where: *r_p_* and *r_s_* are complex reflection amplitude coefficients with parallel (subscript *p*) and perpendicular (subscript *s*) light polarization to the plane of incidence, respectively. The angles Ψ and Δ are referred to as ellipsometric angels. The ellipsometric angles depend on the films, substate, cover parameters and wavelength *λ*.

At the stage of technological research, the refractive index and thickness of composite SiO_x_:TiO_y_ films were determined with the aid of a monochromatic ellipsometer SENTECH SE400 (Sentech, model 2003, *λ* = 632.8 nm, Berlin, Germany). However, to determine the dispersion characteristics of the refractive index and extinction coefficient, the spectroscopic ellipsometer Woollam M2000 (J.A. Woollam Co. Inc., Lincoln, NE, USA) and CompleteEASE software were used.

In ellipsometric spectroscopic measurements, the key issue involves the correct choice of the dispersion model of dielectric functions. When testing various dispersion models, we selected the Tauc–Lorentz [51] dispersion model, defining our films in the best way, which describes well various amorphous materials, such as semiconductors [52], insulators [53], polymers [54], or even organic materials [55]. The Tauc–Lorentz model calculates the imaginary part of the dielectric function *ε**^(i)^* by multiplying the Tauc joint density of states [56] and the $\varepsilon_{L}^{\left(i\right)}$ obtained from the Lorentz oscillator model. In our case, we used the two-oscillator Lorentz model, and hence the expression for the imaginary part of the dielectric function *ε**^(i)^* has the form:(10)$$\varepsilon^{\left(i\right)}=\{\begin{array}{cc} \sum_{j=1}^{2}\frac{A_{j}\cdot E_{j}\cdot C_{j}\cdot {\left(E-E_{g}\right)}^{2}}{{\left(E^{2}-E_{j}^{2}\right)}^{2}+C_{j}^{2}\cdot E^{2}}\times \frac{1}{E} & \begin{array}{ccc} & for & E>E_{g} \end{array}\\ 0 & \begin{array}{ccc} & for & E\leq E_{g} \end{array} \end{array}$$

The seven fitting parameters are *E_g_*, *A*_1,2_, *E*_1,__2_, and *C*_1,2_, and all are in units of energy. The real part of the dielectric function*ε**^(r)^* is derived by Kramers–Kronig integration:(11)$$\varepsilon^{\left(r\right)}\left(E\right)=\varepsilon^{\left(r\right)}\left(\infty \right)+\sum_{j=1}^{2}\frac{2}{\pi }\cdot P_{c}\cdot \int_{E_{g}}^{\infty }\frac{\xi \cdot \varepsilon_{j}^{\left(i\right)}(\xi )}{\xi^{2}-E^{2}}d\xi$$
where the *P_c_* stands for the Cauchy principal part of the integral and eight fitting parameters $\varepsilon^{\left(r\right)}\left(\infty \right)$ is the high frequency dielectric constant. Deriving these integral yields, the analytical expression of the real part of the dielectric function [51]. Refractive index and extinction coefficient have been calculated from the formulas:(12)$$n\left(E\right)={\left[\frac{1}{2}{\left(\varepsilon^{\left(r\right)}{}^{2}+\varepsilon^{\left(i\right)}{}^{2}\right)}^{1/2}+\varepsilon^{\left(r\right)}\right]}^{1/2}$$
and
(13)$$\kappa \left(E\right)={\left[\frac{1}{2}{\left(\varepsilon^{\left(r\right)}{}^{2}+\varepsilon^{\left(i\right)}{}^{2}\right)}^{1/2}-\varepsilon^{\left(r\right)}\right]}^{1/2}$$

Knowing the refractive index n of the material of the fabricated SiO_x_:TiO_y_ composite films and the refractive indexes of the constituent materials (*n_SiO_*_2_ = 1.46, *n_TiO_*_2_ = 2.52), the porosity P of the film material was estimated using the Lorentz–Lorenz formula
(14)$$P=\left[1-\frac{n^{2}-1}{n^{2}+2}\cdot \frac{n_{m}^{2}+2}{n_{m}^{2}-1}\right]\cdot 100\%$$
where *n* is the refractive index of the films obtained from ellipsometric measurements (*λ* = 632.8 nm), and *n_m_* is the refractive index of the two-component film SiO_x_:TiO_y_ calculated from the Lorentz–Lorenz formula based on the known composition of the film material:(15)$$\frac{n_{m}^{2}-1}{n_{m}^{2}+2}=f_{1}\cdot \frac{n_{1}^{2}-1}{n_{1}^{2}+2}+f_{2}\cdot \frac{n_{2}^{2}-1}{n_{2}^{2}+2}$$

We assumed here that *n*_1_º*n**_SiOx_*»*n**_SiO_*_2_ and *n*_2_º*n**_TiOy_*»*n**_TiO_*_2_. The *f*_1_ and *f*_2_ are the molar ratio of each component, which equals 0.34 and 0.66, respectively.

### 3.5. Surface Morphology

The surface morphology of the silica–titania waveguide films was analyzed using a SEM and AFM. The SEM method was used to assess surface smoothness and to examine their cross-sections. The AFM method was used to record the images of film surfaces, and on their basis surface roughness of the films was determined (root mean square roughness).

The measurements of scanning electron microscopy were made with the SEM Supra 35 (Zeiss, Oberkochen, Germany) in the in-lens mode with accelerating voltages in the range from 2 to 10 kV. AFM measurements were carried out using AFM N_TEGRA (NT-MDT, Moscow, Russia). Measurements were carried out at a resonance frequency equal to 136.281 kHz in semi-contact mode with HA_NC (NT-MDT) silicon cantilever with nominal curvature radius of a tip 10 nm. The average roughness measurements (RMS) were evaluated using NOVA 1.0.26.1644 (NT-MTD) software. The images were taken from an area of 1 × 1 μm^2^.

## 4. Results and Discussion

### 4.1. FTIR

The recorded FTIR spectra for the sol and for the powder obtained from it are presented in Figure 4a,b, respectively. The powder was made from the same sol after drying and annealing it at 500 °C. The FTIR spectrum of the sol (Figure 4a) contains significantly more absorption bands than the FTIR spectrum of the SiO_x_:TiO_y_ powder. It complies with the expectations since the sol, also containing solvents, is a more complex chemical composition than the powder. The deep absorption bands in the transmittance of sol at the frequencies of 1044 cm^−^^1^ and 1086 cm^−^^1^, respectively, can come from the silica network and they can be attributed to asymmetric Si-O-Si stretching vibrations. At similar frequencies, there may also occur absorption bands connected with the presence of C-O stretching vibrations, which in the case of sols are a natural consequence of the presence of organic compounds. The absorption band at 802 cm^−^^1^ may result from symmetrical Si-O-Si vibrations and symmetric vibrations of oxygen atoms parallel to the Si-O-Si bond [32,57]. The band at 430 cm^−^^1^ corresponds to Ti-O bonds in the TiO_2_ structure [32]. The wide absorption band at the frequency of 548 cm^−^^1^ may be related to the Si-C stretching vibrations, which indicates the presence of chemical bonds between the organic and inorganic components of the sol [58]. The same frequency (548 cm^−^^1^) is assigned to the absorption band of the Ti-O-Ti bonds [57]. The absorption band at 952 cm^−^^1^ is assigned to both the stretching vibrations of Si-O-Ti and the vibrations from SiO- and Si-OH groups [32,59]. From hydroxyl groups with hydrogen bonds also come the absorption bands at the frequencies of 3299 cm^−^^1^ (stretching vibrations), 1656 cm^−^^1^ (bending vibrations of water), 1380 cm^−^^1^ (deformation vibrations) and 879 cm^−^^1^ (vibrations of the -OH groups with methyl groups) [45,60]. The remaining absorption bands at the frequencies of 2972 cm^−^^1^ and 2880 cm^−^^1^ are attributed to the aliphatic C-H stretching vibrations.

In the FTIR spectrum of the SiO_x_:TiO_y_ powder (Figure 4b) there are no absorption bands resulting from the vibrations of hydroxyl groups (3299 cm^−^^1^, 1380 cm^−^^1^). Additionally, we can observe a weak absorption band at the frequency of 1620 cm^−^^1^, which probably comes from the residual vibrations of the -OH groups, which is the result of water adsorption by SiO_2_, as silica has a high ability to adsorb it [60]. After annealing of the SiO_x_:TiO_y_ powder at the temperature above 400 °C, no absorption bands coming from organic groups were recorded. In the frequency range from 400 cm^−^^1^ to about 1200 cm^−^^1^, we observe vibration bands of the Si-O and Ti-O bonds. There is a characteristic absorption band coming from the asymmetric stretching vibrations of Si-O-Si (1030 cm^−^^1^), which in the spectrum of the sol could be obscured by the stretching vibrations band of C-O. The presence of the absorption band at the frequency of 913 cm^−^^1^ in the solid phase, resulting from the vibrations of the Si-O-Ti cross-bonds, is interpreted as evidence confirming the presence of the amorphous phase [32].

### 4.2. Raman Spectroscopy

Baseline correction for all data was conducted prior to the analyzing procedure and each spectrum was normalized. Figure 5 shows the Raman spectra of a raw xerogel sample (obtained at room temperature) and sample annealed at 500 °C for 60 min. For the purpose of accurate analysis in all the spectra obtained, the spectral range was divided into two regions. As we can see, the Raman spectrum for the raw sample contains peaks that are characteristic of both the post-process organic phase and the titanium oxide. The former phase is manifested by the presence of bands in the high wavenumber range, where the stretching transition of water is observed in the range 3000–3600 cm^−1^ which comes from both symmetrical and asymmetrical stretching vibrations of the –OH group [61]. Moreover, we see a bending overtone of water centered around 1640 cm^−1^. The weak peak around 2900 cm^−1^ comes from the C–H stretching vibrations. This peak is characteristic for ethanol. In the low wavenumbers range, we observed broad bands with the wide line width located at 930 cm^−1^, 830 cm^−1^, 620 cm^−1^, 506 cm^−1^, 400 cm^−1,^ 255 cm^−1^ and 148 cm^−1^. The peaks are observed in the vicinity of the active modes of TiO_2_ (Table 1 in Ref. [62]). However, the weak Raman signal corresponds to the low degree of crystallinity indicating the dominance of amorphous phase. The spectrum of the raw sample also shows a weak band around 1075 cm^−1^. This band can be derived from asymmetric and symmetric stretching vibrations of the silica network [63].

This band, however, is not noticeable for the sample annealed at 500 °C. In general, silica gives a weak Raman signal, and when anatase bands clearly develop in the spectrum (at high temperature), the silica signal may be weakened and covered with stronger titanium oxide lines. In a Raman spectrum of a sample annealed at 500 °C, the peaks from the organic phase disappeared. Moreover, we can see that the peaks located at 634 cm^−1^, 512 cm^−1^, 390 cm^−1^ and 144 cm^−1^ are well developed. The broad peaks observed previously become sharper and more intense, signifying an increase in crystalline phase content in the sample. The observed peaks are consistent with the presence of the anatase phase. Especially, the most intense band centered at 144 cm^−1^ is a particularly sensitive detector of even very small amounts of crystalline TiO_2_ in anatase form. In general, anatase has the space group D4h (I41/amd). Based on the factor group analysis, and assuming site symmetries for Ti and O atoms in the unit cell, there are six Raman active modes for anatase, assigned as follows: A1g (512 cm^−1^) + B1g (399 cm^−1^) + B1g (518 cm^−1^) + Eg (144 cm^−1^) + Eg (198 cm^−1^) + Eg (639 cm^−1^). The weak band observed in spectrum at about 830 cm^−1^ was assigned as the first overtone of the B1g mode of anatase.

### 4.3. Effect of Withdrawal Speed

Exemplary dependences of the final thicknesses d and refractive indexes n of the composite films SiO_x_:TiO_y_ on the withdrawal speed v are presented in Figure 6. Full squares are used to indicate the experimental dependence *d* = *d*(*v*), and empty squares are used to present the experimental dependence *n* = *n*(*v*). The film thicknesses increase with the increasing speed *v*, while the refractive indices remain constant. By changing the speed *v* from 3.9 to 6.5 cm/min, we obtained SiO_x_:TiO_y_ films with thicknesses ranging from 130 to 170 nm. The experimental dependence *d* = *d*(*v*) was approximated by the function (5), where *a* = 0.522 ± 0.002 and *A* = (63.70 ± 0.15) nm. Additionally, the experimental dependence of the refractive index *n* on the withdrawal speed was approximated by the linear function *n* = *n*_0_ + *g*×*v*, obtaining *n*_0_ = 1.9372 ± 0.0012 and *g* = (−0.0012 ± 0.0002) min/cm. Thus, we can observe that the refractive index of the fabricated films depends to a very small extent on the withdrawal speed *v*. In both cases, the experimental points are close to the approximating lines.

### 4.4. Effect of Sol Aging

The hydrolysis and condensation reactions are still taking place in the sol, even when it is stored at lower temperature. Thus, the sol thickens over time and its viscosity increases. This results in increasingly thicker films, produced at the same withdrawal speed *v*. This effect is illustrated in Figure 7, which shows, respectively, the dependence of the final film thickness and its refractive index on the storage time of the sol, for the withdrawal speed *v* = 5.0 cm/min. Everytime sols after usage were stored at a temperature of 4 °C. The points marked with diamonds and the error bars were determined from the approximation of the experimental characteristics *d* = *d*(*v*) and *n* = *n*(*v*) (Figure 4). Here, maximum uncertainties are plotted with error bars. It can be seen that the final film thickness increases linearly with the storage time of the sol, and the refractive index remains constant at ~1.94. This means that the produced sol can be used for a longer period of time, yet it requires that we know the influence of its storage time on its viscosity and, consequently, on the thickness of the produced films at specific withdrawal speeds. The knowledge of such dependencies as shown in Figure 7 is necessary from the viewpoint of the designing processes for the fabrication of waveguide films.

### 4.5. Surface Morphology

Figure 8a presents an exemplary, AFM image of the surface of a SiO_x_:TiO_y_ composite film deposited on a soda-lime glass substrate, and after heat treatment for 1 h at 500 °C. As we can easily observe, the film is very smooth, and the difference between the lowest and the highest point on the tested surface 1.0 × 1.0 μm^2^ is 1.19 nm, while the root mean square surface roughness derived from this image is *rms* = 0.15 nm. Section 4.6 shows waveguide properties of the SiO_x_:TiO_y_ composite films presented here. The roughness of the interphases of the waveguide film is the source of scattering losses. These losses are proportional to the squared *rms*, to ${\left(n_{1}^{2}-n_{c}^{2}\right)}^{2}$ and to the power density on the boundary surfaces of the waveguide film [64]. In single-mode waveguide films of high refractive index, there are high power densities on the boundary surfaces, and hence the acquisition of low scattering losses is conditioned by their low roughness. In binary composite waveguide films, SiO_x_:TiO_y_ containing more than 20% of titanium oxide, it tends to crystallize and form a separated crystal phase, which negatively affects the smoothness of the film surface. In such a case, an increase in scattering losses on the surface of the waveguide film will be observed, and strong scattering in the volume of the film material will occur. We have to face a very difficult technological challenge to prevent this effect. High smoothness of the surface can be interpreted as a confirmation of the amorphous nature of the SiO_x_:TiO_y_ composite waveguide films presented here.

The SEM image of the upper surface of the waveguide film is shown in Figure 9a, while the cross-section of the structure is shown in Figure 9b. In Figure 9a, a smooth, continuous surface of the composite waveguide film is observed, with no cracks or visible defects. Figure 9b, which shows a cross-section of the film applied onto soda-lime glass substrate, indicates a uniform film thickness (*d* = 159.2 nm). SEM images confirm high homogeneity of the SiO_x_:TiO_y_ films and high smoothness of their surfaces.

### 4.6. Optical Properties

The ellipsometric spectroscopic studies were performed for the SiO_x_:TiO_y_ composite film fabricated on a silica substrate whose optical dispersion properties are very well known [65].

The registered dispersion characteristics of the ellipsometric angles *Ψ*(*λ*)and Δ(*λ*) at the illumination angle of the sample *θ* = 70° were plotted with solid lines in Figure 10. The modeled dispersion characteristics were plotted with dashed lines, calculated using the Tauc–Lorentz formula presented in Section 3.4, in which two oscillators were taken into account. For the applied model, eight parameters were fitted, whereof the calculated values are summarized in Table 1. As we can observe in Figure 10, a perfect fit of the modeled dispersion characteristics of the ellipsometric angles to the experimental characteristics was obtained. The calculations were used to determine the thickness of the SiO_x_:TiO_y_ composite film *d* = 139.2 nm. The refractive index *n*(*l*) and the extinction coefficient *k* (*l*), determined, respectively, from the formulas (12) and (13), are shown in Figure 11 (*E* = 1240/*l*). As we can observe, in the spectral range above 300 nm, normal dispersion of the refractive index occurs, while below the wavelength of 300 nm, anomalous dispersion is observed, which is related to the presence of an absorption band in this spectral range.

Optical absorption in composite SiO_x_:TiO_y_ films was studied by UV-Vis spectroscopy in the spectral range of 200–1100 nm. The plots of transmittance of the selected composite film SiO_x_:TiO_y_ of different thicknesses are shown in Figure 12. In this Figure is shown the transmission characteristic of the soda-lime glass substrate (depicted as a grey line). The spectra of SiO_x_:TiO_y_ films show a high transmission above 400 nm wavelength and a strong absorption edge (observed as a decrease in transmittance) below 400 nm.

The line of transmittance of the soda-lime slide is the upper envelope of transmittance of composite SiO_x_:TiO_y_ films which indicate that those films are uniform. Additionally, the uniformity of SiO_x_:TiO_y_ films is indicated by the small refractive index variation, which was mentioned in Section 3.2.

The plots of reflectance characteristics of selected composite SiO_x_:TiO_y_ films are also shown in Figure 12. The solid grey line is corresponding to the reflectance of the soda-lime glass substrate. As can be seen, that interference minima lie on the reflectance characteristic of the soda-lime glass substrate for the wavelengths above 400 nm, which clearly shows the homogeneity of the layers and the lack of absorption.

Based on the UV-Vis transmittance spectra, the optical band gap was determined using Tauc’s Equation (8). Figure 13 shows the plot of ${\left(\alpha \cdot h\nu \right)}^{2}$ and ${\left(\alpha \cdot h\nu \right)}^{1/2}$ on the energy photon for two selected silica–titania layers with different thicknesses, coated on a soda-lime glass substrate. The optical bandgap was calculated by extrapolating the linear part of the curve as a function of (hν) curve to α = 0. For these layers, the determined values of the indirect energy bandgaps were 3.589 eV and 3.543 eV for the thicknesses of 145 and 166 nm, respectively (Figure 13a), whereas the direct energy bandgaps were 4.012 eV and 3.985 eV for 145 and 166 nm, respectively (Figure 13b). The high values of energy band gaps proved the amorphous structure of the presented SiO_x_:TiO_y_ waveguide layers.

Table 2 shows the values of energy bandgaps for the silica–titania layers obtained at various time intervals. As can be seen, the value of the energy bandgap practically does not depend on the aging time of the sol.

Taking into account the refractive index of silica *n_SiO_*_2_ = 1.46 and titania *n_TiO_*_2_ = 2.52, using formula (15), we obtained *n_m_* = 2.05, and then using the measured values of the refractive index *n* of the produced composite SiO_x_:TiO_y_ films, their volume percentage porosity *P* was determined using the Lorentz–Lorenz formula (Equation (14)). We found that the porosity of the composite SiO_x_:TiO_y_ fabricated by us is P ≈ 7% for all the prepared samples.

### 4.7. Waveguide Properties

The SiO_x_:TiO_y_ composite films presented here have good waveguide properties. Figure 14 shows the images of an excited waveguide into which light was launched with the use of a prism coupler and by means of the optical tunneling effect. We can observe streaks of scattered light whereof intensity decreases along the path of propagation. It can be seen from the comparison of the images for TM_0_ and TE_0_ modes that stronger optical losses take place for TE polarization. From the light intensity distribution along the streak, the optical loss factor *m* can be determined. It is assumed in this procedure that: (*i*) the waveguide is homogeneous in the direction of light propagation, (*ii*) at each point *x* of the scattered light streak, its intensity is proportional to the optical power that is guided in the waveguide film at this point. With such assumptions, it can be written that the light intensity distribution along the streak is determined by the relationship: *P*(*x*) = *P*(*x = 0)**⋅* exp(-*m×**x)*. For the waveguide shown in Figure 14 (*n*_1_ = 1.9383, *d* = 145.7 nm), the determined optical losses are, respectively, *a**_TM_* = 4.343 *m* = 1.1 dB/cm for mode TM_0_ and *a**_TE_* = 4.343 *m* = 2.7 dB/cm for mode TE_0_ (at *λ* = 676.7 nm). These losses are several times higher than those for the previously reported SiO_x_:TiO_y_ waveguide films with lower refractive indices [40,42]. This result does not come as a surprise to us, because in the composite SiO_x_:TiO_y_ films presented here we have a much higher content of titanium. Similar optical losses were obtained by Heideman et al. [25] for ZnO waveguide films with refractive indices of 1.93–1.96 (at *λ* = 632.8 nm), which were produced by the RF magnetron sputtering method. Ref. [28] reported erbium-doped ZrO_2_ waveguide films with refractive indices of 1.83–1.92 and optical losses exceeding 1 dB/cm. In the waveguide film TiO_2_ of refractive index *n* ~2 and thickness *d* = 336 nm, Tauam et al. [22] measured the optical losses for the fundamental mode TM_0_ amounting to 0.8 dB/cm. However, it should be emphasized that it was a two-mode waveguide, and the measurement was made only for the fundamental mode. In the measurements of our films by spectroscopic ellipsometry and spectrophotometric measurements, we did not report the presence of absorption bands in the Vis-NIR spectral ranges above the wavelength of 400 nm. This means that the optical losses can only be affected by scattering on the boundary surfaces of the waveguide film and on the inhomogeneities of the material in its volume. The AFM and SEM tests demonstrated high surface smoothness of the waveguide film, and hence it can be concluded that its influence on optical losses is negligible. Thus, the main source of losses may be attributed to the scattering on the surface waveguide film/substrate and to the scattering in the volume of the film.

Figure 15 presents mode characteristics of a slab waveguide with the refractive indices of *n_b_*-*n*_1_-*n*_c_ = 1.51-1.94-1.00, calculated for the wavelength *λ* = 677 nm. The cut-off thicknesses are 66.75 nm and 134.25 nm for the base modes TE_0_ and TM_0_, and 344.25 nm and 456.00 nm for the first order modes TE_1_ and TM_1_, respectively. Figure 16 presents the normalized power density distributions of the fundamental modes TE_0_ and TM_0_ calculated with the use of the transfer matrix method [38]. The area of the waveguide film is marked in gray. As we can observe, power density of the fundamental mode TE_0_ is greater than that of mode TM_0_ both on boundary surfaces of the waveguide film and in its volume. Hence, for the TE_0_ mode, light scattering both on boundary surfaces of the waveguide film and in its volume are greater than that for the TM_0_ mode. Consequently, as can be seen from Figure 14, optical losses for the TE_0_ mode are higher than those for the TM_0_ mode.

After the application of a sol film on the soda-lime substrate, a grating coupler of period of *Λ* = 417 nm was fabricated in it using the nanoimprint method, and then the structure was subjected to annealing at the temperature of 500 °C for 60 min. In this way, a waveguide structure with a grating coupler was obtained, which allows for efficient light coupling into the waveguide film. The sol film was applied at the same withdrawal speed as for the structure shown in Figure 14. As demonstrated in Refs. [38,42], planar waveguide structures with grating couplers are highly sensitive sensor structures. Figure 17 shows a planar waveguide structure, excited with the use of the fabricated grating coupler. We can observe a streak of scattered light along the waveguide and a very strong optical signal that reached the matte area on the applied glass substrate. Figure 18 shows the excitation characteristics of a planar waveguide structure with a grating coupler, recorded in the measurement system described in Ref. [42]. These characteristics are symmetrical to the normal. The fundamental mode TM_0_ was excited at the synchronous angles of *q**_TM_* = ±5.63887°, while the excitation of the TE_0_ mode was carried out at the synchronous angle of *q**_T_**_E_* = ±0.24499°. The angles were used to calculate effective refractive indices (Equation (3), Ref. [42]) and then the thickness of the waveguide film *d* = 144.8 nm and the refractive index *n*_1_ = 1.9382 were calculated from the characteristic equations (Equation (6), Ref. [42]). This calculation procedure is based on the assumption that the waveguide film is homogeneous. We infer its homogeneity based on its reflectance spectrum as is described in Section 4.6. As can be observed, the parameters of the waveguide film with a grating coupler are almost identical as those of the film shown in Figure 14. The use of the waveguide films presented here in sensor structures with grating couplers will allow us to achieve high optical sensitivities. The parameters of the waveguide film of the structure shown in Figure 17 are close to the optimal ones in terms of optical sensitivity [42].

## 5. Conclusions

The paper presents the results of the characterization of SiO_x_:TiO_y_ composite films fabricated by the sol–gel method and dip-coating technique on soda-lime glass substrates. The studies involving the impact of aging time of the sol demonstrated its impact on viscosity and thus on the final thickness of the films produced at the same withdrawal speed. Yet, the aging of the sol does not affect the final refractive index. The AFM and SEM tests demonstrated high surface smoothness of the produced SiO_x_:TiO_y_ films, while the spectrophotometric tests demonstrated their optical homogeneity. The studies of optical energy band gaps by the Tauc method did not display any influence of sol aging time on them. High values of the optical band gaps indicate the amorphous nature of the produced films. This was confirmed by the results of FTIR studies. Fourier spectroscopy studies showed the absence of organic components in the layers after annealing at the temperature of 500 °C. Spectrophotometric tests and the tests with the use of spectroscopic ellipsometry did not indicate the presence of absorption bands in the spectral range above 400 nm. The determined losses for the film thickness slightly above the cut-off thickness are, respectively, 1.1 dB/cm for the TM_0_ mode and 2.7 dB/cm for the TE_0_ mode (at wavelength λ = 676.7 nm). We demonstrated a structure with the fabricated grating coupler of the period of *Λ* = 417 nm, which can be applied as a sensor structure. The high refractive index of the waveguide film means that high optical sensitivities can be achieved. The films presented in this paper may become a technological platform for highly sensitive evanescent wave chemical/biochemical sensors in the future. We believe that the use of glass substrates of better surface quality will allow us to obtain lower optical losses.

## Figures and Tables

**Figure 1 materials-14-07125-f001:**
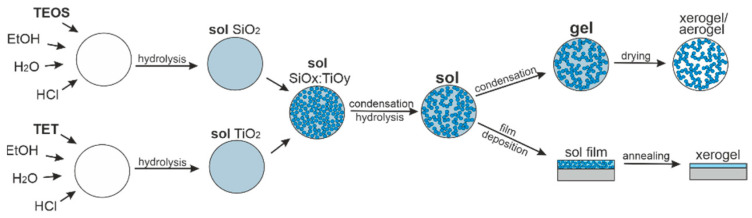
Scheme of sol–gel processes.

**Figure 2 materials-14-07125-f002:**
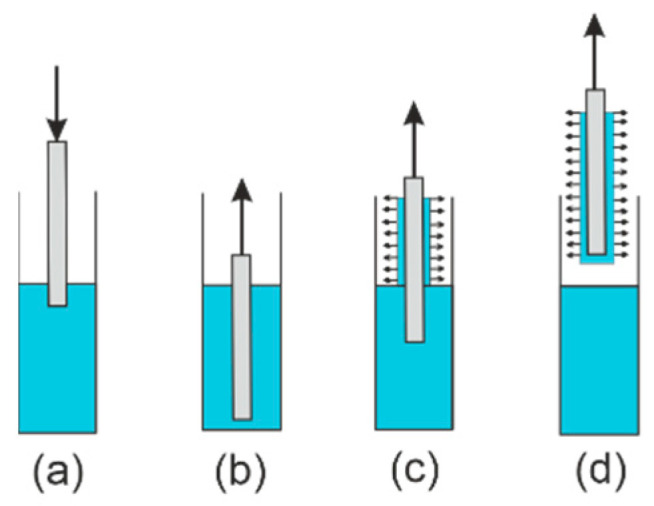
The stages of dip-coating technique. (**a**) Dipping the substrate in the sol, (**b**) beginning of the withdrawal, (**c**) withdrawal of the substrate from the sol, (**d**) pre-drying the layer.

**Figure 3 materials-14-07125-f003:**
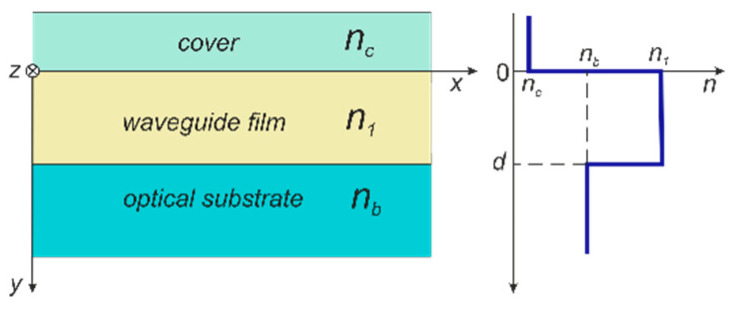
Scheme of the optical slab waveguide and its refractive index profile.

**Figure 4 materials-14-07125-f004:**
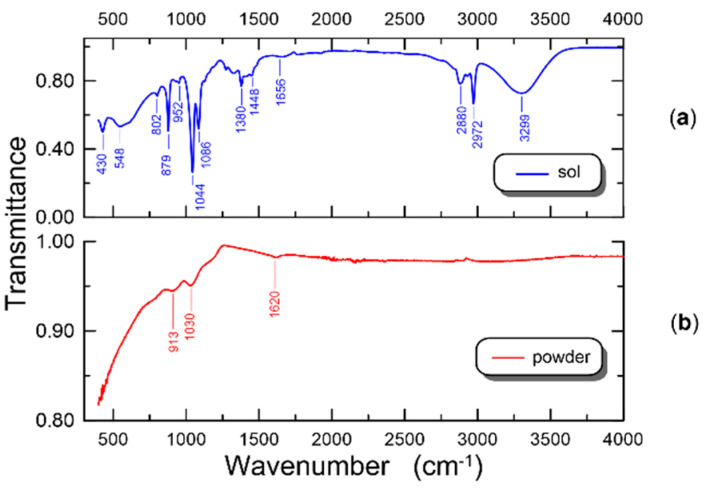
FTIR spectrum of the sol (**a**) and powder of silica-titania (**b**).

**Figure 5 materials-14-07125-f005:**
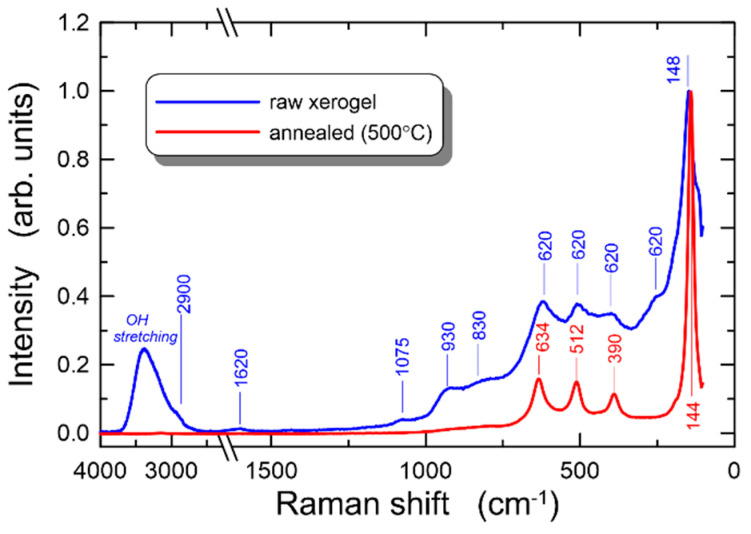
Comparison of Raman spectra for raw xerogel and xerogel annealed at 500 °C for 60 min.

**Figure 6 materials-14-07125-f006:**
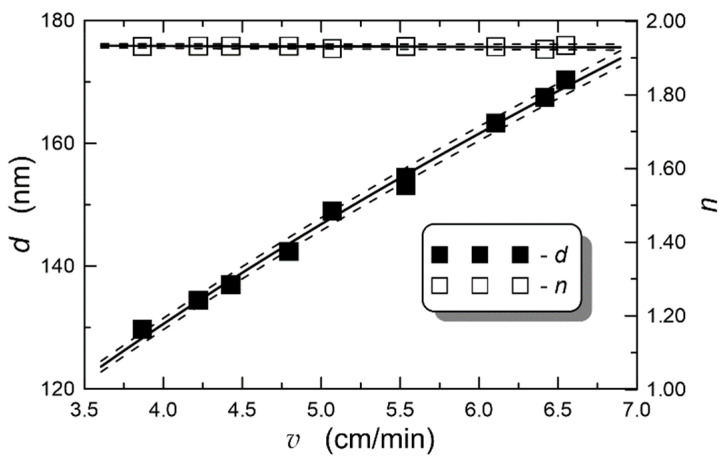
Influence of substrate withdrawal speed from sol on final thickness and refractive index of the composite SiO_x_:TiO_y_ films. Aging time of sol equal 27 days, annealing at 500 °C for 1 h. *λ* = 632.8 nm.

**Figure 7 materials-14-07125-f007:**
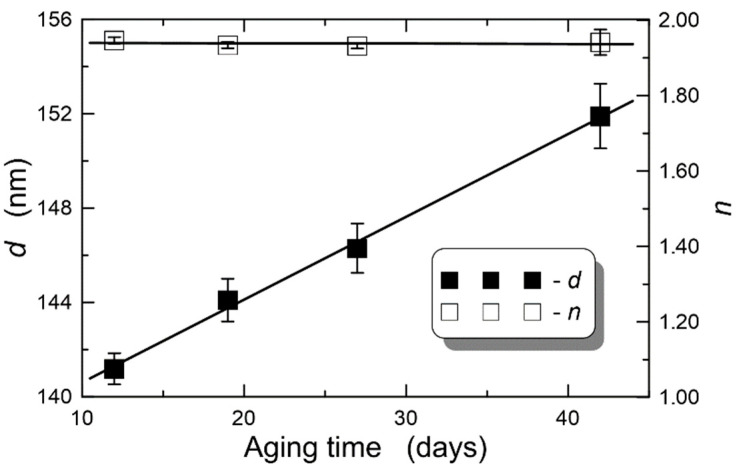
Influence of sol aging time on film thickness and refractive index for withdrawal speed of substrates from sol *v* = 5.0 cm/min. *λ* = 632.8 nm.

**Figure 8 materials-14-07125-f008:**
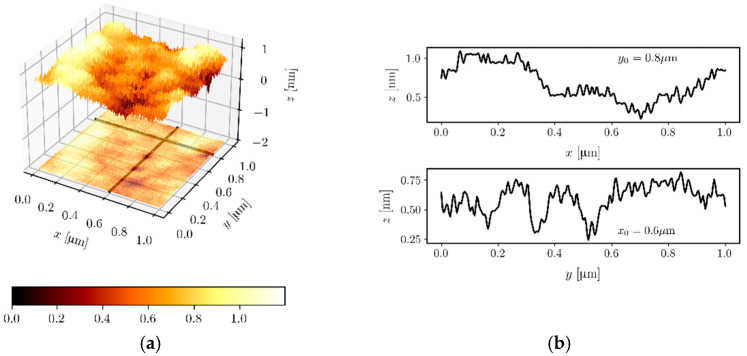
(**a**) AFM topographical image of an SiO_x_:TiO_y_ thin film deposited on a soda-lime glass substrate, obtained for the area of size 1.0 × 1.0 μm^2^. (**b**) Surface profiles in two cross-sections.

**Figure 9 materials-14-07125-f009:**
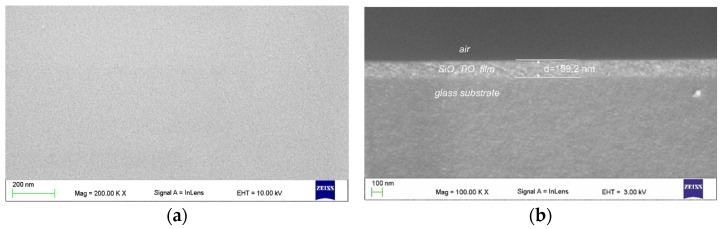
SEM images of top (**a**) and cross section (**b**) of the slab waveguide structure.

**Figure 10 materials-14-07125-f010:**
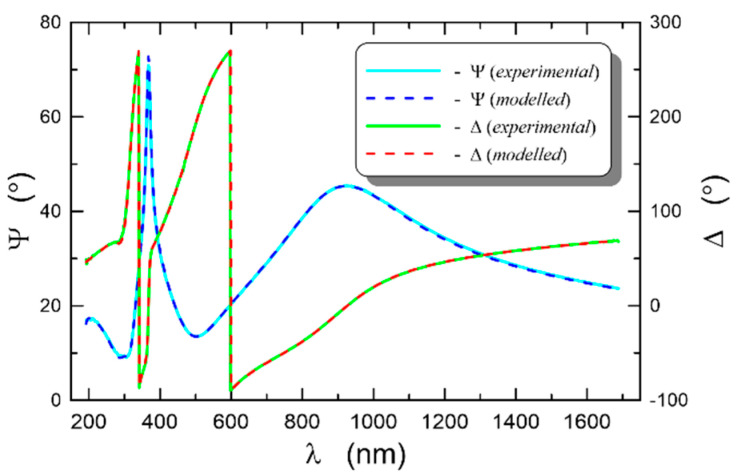
Dispersion characteristics of the ellipsometric angles for composite a SiO_x_:TiO_y_ film on a silicon substrate. The illumination angle equals 70°.

**Figure 11 materials-14-07125-f011:**
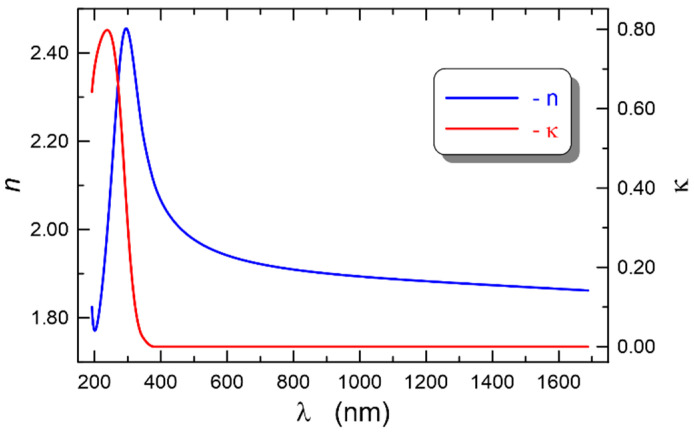
Dispersion characteristics of the refractive index and extinction coefficient for a composite SiO_x_:TiO_y_ film.

**Figure 12 materials-14-07125-f012:**
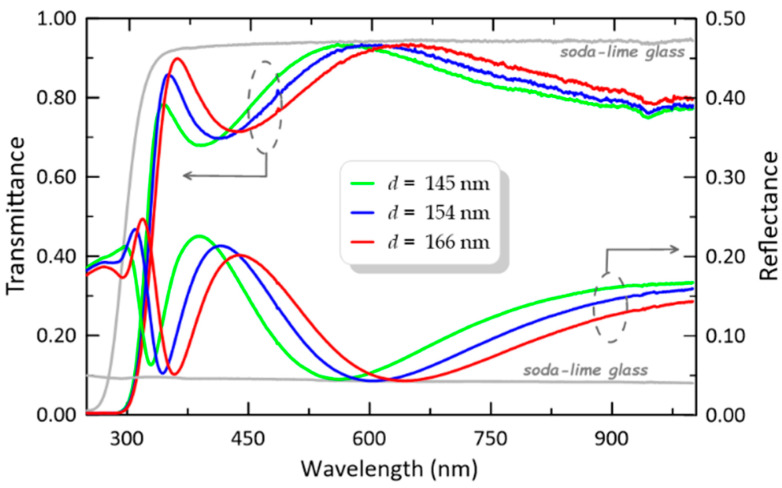
Transmittance and reflectance spectra of the SiO_x_:TiO_y_ films for selected thickness.

**Figure 13 materials-14-07125-f013:**
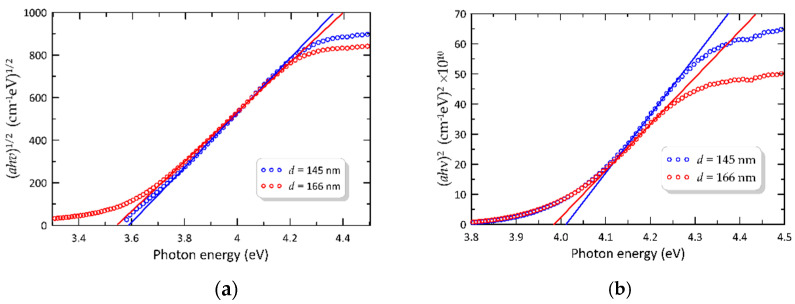
Tauc plot of ${\left(\alpha \cdot h\nu \right)}^{1/2}$ and ${\left(\alpha \cdot h\nu \right)}^{2}$ as a function of (hν) for indirect (**a**) and direct (**b**) optical transitions, respectively.

**Figure 14 materials-14-07125-f014:**
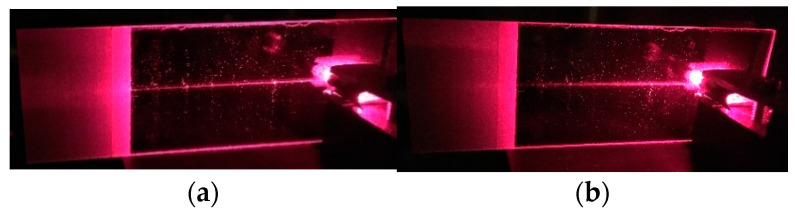
Images of the excited slab waveguides. (**a**) TM_0_ mode, (**b**) TE_0_ mode, *λ* = 676.7 nm.

**Figure 15 materials-14-07125-f015:**
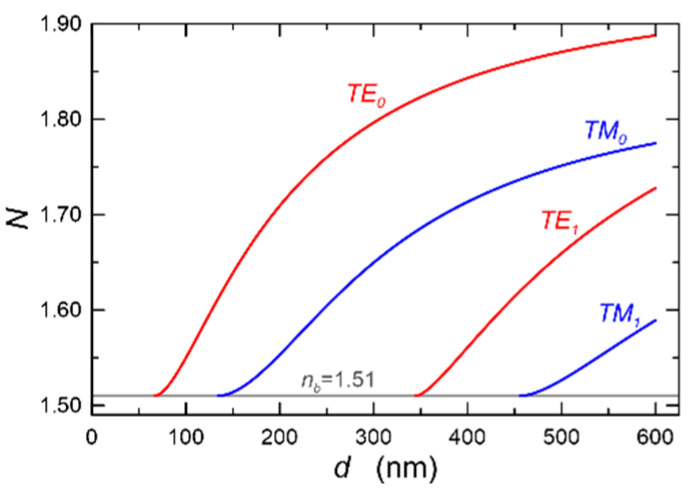
Modal characteristics of the slab waveguide. *n_b_-n*_1_*-n_c_* = 1.51-1.94-1.00, λ = 677 nm.

**Figure 16 materials-14-07125-f016:**
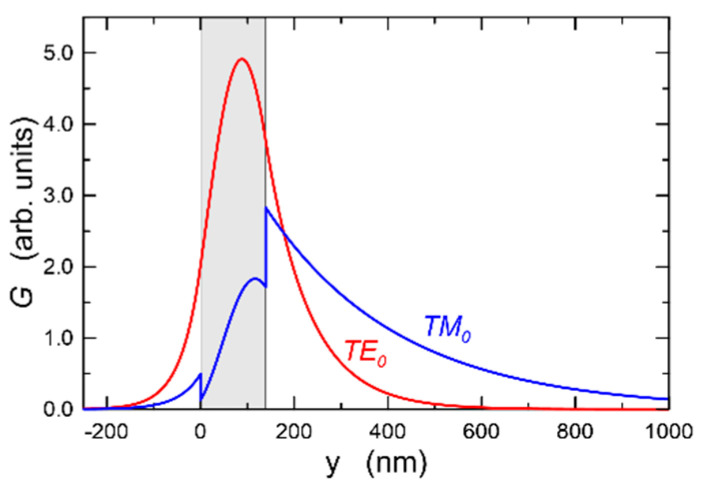
Distribution of the power density in the slab waveguide *n_b_-n*_1_*-n_c_* = 1.51-1.94-1.00, *d* = 146 nm, λ = 677 nm.

**Figure 17 materials-14-07125-f017:**
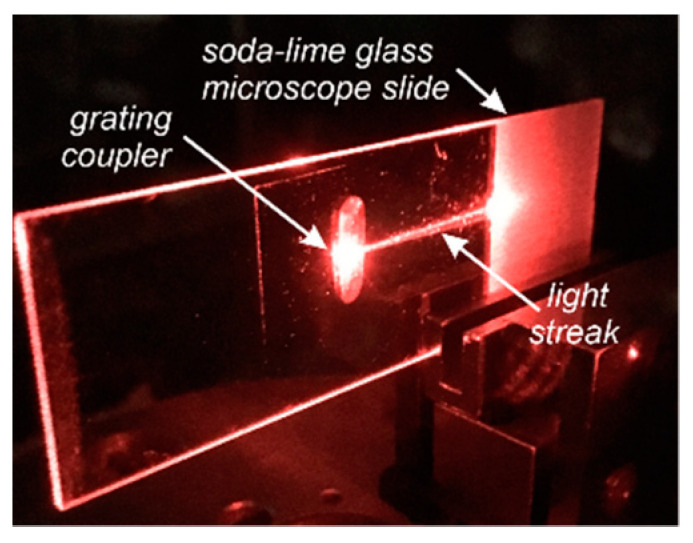
Slab waveguide excited with the use of grating coupler. λ = 677 nm, Λ = 417 nm.

**Figure 18 materials-14-07125-f018:**
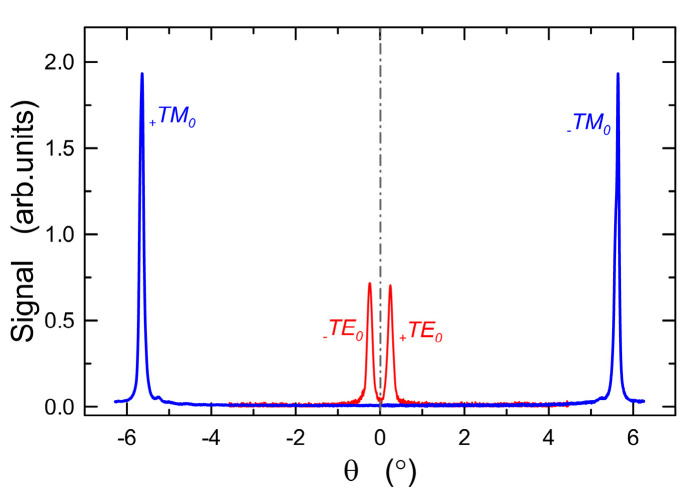
Modal spectra of the slab waveguide excited with the use of the grating coupler for fundamental modes, λ = 677 nm, Λ = 417 nm.

**Table 1 materials-14-07125-t001:** Parameters of the Tauc–Lorentz model fit to experimental data.

*E*_*g*1_ (eV)	*A*_1_ (eV)	*E*_01_ (eV)	*B*_1_ (eV)	*E*_*g*2_ (eV)	*A*_2_ (eV)	*E*_02_ (eV)	*B*_2_ (eV)
3.323 ± 0.005	93 ± 4	6.2 ± 0.3	14± 1	33.600 ± 0.006	93.5 ± 2.3	4.28 ± 0.01	1.50 ± 0.02

**Table 2 materials-14-07125-t002:** Influence of sol aging time on optical band gaps.

Aging Time (Day)	*d* (nm)	*E_g_* Indirect (eV)	*E_g_* Direct (eV)
19	165.13	3.580	3.973
27	148.89	3.618	4.021
	170.36	3.591	4.006
42	160.63	3.589	4.012
	177.42	3.590	3.983

## Data Availability

Data sharing is not applicable to this article.

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
