# Peer review of "High Refractive Index Silica-Titania Films Fabricated via the Sol–Gel Method and Dip-Coating Technique—Physical and Chemical Characterization"

_materials, 2021, doi:10.3390/ma14237125_

Round 1

Reviewer 1 Report

  1. For a complete understanding of sol-gel issues, I recommend Chapter 2.1. insert hydrolysis and condensation reactions for SiO2 and TiO2.
  2. In Chapter 2.2, I recommend entering the temperature of the sol preparation and the mixing time. I also recommend inserting the aging time of the sol into this chapter.
  3. Was concentrated HCl used in the sol preparation? If HCl solution has been used, I recommend entering its concentration.
  4. The final molar ratio of sol does not indicate the amount of HCl used.
  5. What was the dwell time of the glass substrate in the sol during the dip-coating process?
  6. What was the volume of the sol into which the glass substrates were immersed? The ration S/V value is important when forming coatings (S – surface of substrate, V – volume of sol).
  7. Was the viscosity of the sol measured after each day of aging (19, 27, 42 days)?

Author Response

We are grateful to the Reviewer for valuable comments and remarks. They helped us to improve the manuscript. Our answers are in the attached file Response_to_Reviewer_1.docx

Reviewer 2 Report

The proposed paper describes the elaboration and the characterization of silica-titania films obtained from the sol-gel way and by dip-coating technique. These films could have an interest for some applications due to their waveguide properties. 

The manuscript is well structured and well written with good quality of figures. The conclusions are clear and adequate to the presented data. Presented results and their analyzes are scientifically sound also I therefore recommend the publish of this work in Materials.

However, some points need to be clarified and completed :

- some references must be added as :

  • E. Bendini et al., J Phys Chem C 2017, 121, 14572-14580 about the dip-coating process ;
  • B. Louis et al., J Phys Chem C 2011, 115, 3115-3122 about the crystallization of anatase into binary SiO2/TiO2 sol-gel thin films ;
  • M. Royon et al., Aerospace, 2021, 8, 109 about application of waveguide.

- the form of references must be improved in particular the chemica formulae.

- in the section 2.3 :

  • the chemical composition of the TET and the TEOS solutions must be given, not only the final sol.
  • why the films are not characterized by FTIR spectroscopy? The spectrum of powder could be slightly different.

-In the abstract, the authors mentioned the refractive index of TiO2-SiO2 films (1.94). I suggest that you can add at which wavelength since refractive index is dependent on this quantity. Can you do the same for the optical losses for TM0 and TE0 modes?

- The waveguiding properties are determined by the refractive index contrast between the sol-gel film and the substrate (soda lime glass). The authors did not give the corresponding refractive index. Can you add this information? I suggest that you can find the data on https://refractiveindex.info/. It should be close to 1.52 for such glass in the visible.

-Regarding the light injection, did you try to inject IR light?

-The films are treated at 500°C where anatase is obtained. Did you try to reach brookite or rutile form and check their impact on the waveguiding properties? Do you think films will crack if brookite or rutile (higher temperature) is reached?

- As I understand, your developed sol-gel is not photosensitive with respect to UV photons for micro or nanostructuration and it is only possible using nanoprinting technique. Do you think it is possible to have a photopatternable one by incorporation of commercial photo-initiator during the sol-gel elaboration or by changing TEOS by another silicate precursor?

- In figure 12, Reflactance should be replaced by Reflectance (Y-label on the right)

-When you give the optical losses in section 4.7 (Waveguide properties), you mention 2 times αTM (1.1 dB/cm and 2.7 dB/cm). I think the second one is αTE.

Author Response

We are grateful to the Reviewer for valuable comments and remarks. They helped us to improve the manuscript. Our answers are in the attached file Response_to_Reviewer_2.docx

Reviewer 3 Report

This article deals with the realization of high refractive index layers for optical applications thanks to the sol gel process. By varying the molar ratio of several metallic oxyde precursors (Si and Ti) authors can tune the optical properties of the final material. Authors have previously worked on 1:1 Si-Ti molar ratio leading to a 1.8 refractive index material.

In this paper, they present results, in terms of optical properties, and homogeneity of a 1:2 molar ratio leading to a 1.94 refractive index layered material. They highlight the quality of this material by using it as optical waveguide.

This study is quiet complete, but following points should be considered:

1- To my mind, it is mandatory, in order to be able to reproduce this study, that authors gives all the details in the section "Sol preparation and film fabrication". Indeed, authors do not give the molar ratio for partial hydrolysis

2- As optical properties are dispersives, authors should give both refractive index and the wavelength.

3- In the comments of figure 13, authors give 3.997 eV for the 166 nm direct energy bandgaps. On the figure it is close to 3.97

4- With the measurement of the two synchronous angles (for TE0 and TM0) optical modes, authors determine two unknowns : thickness and refractive index. This means that they assume an isotropic layer. With such an annealing (500°C) this assumption should be reconsidered

Author Response

We are grateful to the Reviewer for valuable comments and remarks. They helped us to improve the manuscript. Our answers are in the attached file Response_to_Reviewer_3.docx

Reviewer 4 Report

The manuscript presents that the SiOx:TiOy composite films with the refractive index of ~1.94, which are fabricated on soda-lime glass substrates by the sol-gel method and dip-coating technique. In addition, the films demonstrated the good waveguide properties. The logic of the article and the experiment results seemed to be easily understood and comprehensive. Therefore, I recommend the possible publication after some minor revision based on the following comments:

  1. In the introduction, it is better to add the advantages and disadvantages of methods including physical methods, e-beam evaporation, magnetron sputtering, pulse laser deposition, chemical spray pyrolysis, atomic layer deposition, sol-gel methods. Because in the current situation, the innovation of this manuscript was not clearly. It is also strongly suggested to directly point out the technical or scientific challenges in this field.
  2. There are some mistakes that could have been avoided in the context. For example: (a) The case of the first letter of the word, for example, Page 15 line 21 “table 2”, (b) The label of peak positions in the Figure 5 Raman spectra such as the peaks of 506 cm-1, 400 cm-1, and 255 cm-1 were marked wrong as well.
  3. Some figures composed of several picture, please label a, b, c….and give the detailed figure notes. For example, Figure 8, Figure 13 and so on.
  4. The Authors say that thin film technologies have played a key role in the development of microelectronics, and now they dominate also in optoelectronics. I believe it is better to add more references for convenient understanding in this research interdisciplinary area. For example, the references as ACS Appl. Mater. Interfaces,12(2020) 45549, Phys. Lett. 114(2019)253502 etc.

Author Response

We are grateful to the Reviewer for valuable comments and remarks. They helped us to improve the manuscript. Our answers are in the attached file Response_to_Reviewer_4.docx
